# User experience, game satisfaction and engagement with the virtual simulation VR FestLab for alcohol prevention: A quantitative analysis among Danish adolescents

Julie Dalgaard Guldager[1,2]*, Robert Hrynyschyn[3], Satayesh Lavasani Kjær[1], Timo Dietrich[4], Gunver Majgaard[5], Christiane Stock[1,3]

1 Department of Public Health, Unit for Health Promotion Research, University of Southern Denmark, Esbjerg, Denmark, 2 Department of Physiotherapy, University College South Denmark, Esbjerg, Denmark, 3 Institute of Health and Nursing Science, Charité, Universitätsmedizin Berlin, corporate member of Freie Universität Berlin and Humboldt-Universität zu Berlin, Berlin, Germany, 4 Social Marketing @Griffith, Griffith Business School, Griffith University, Gold Coast, Australia, 5 Game Development and Learning Technology, The Maersk Mc-Kinney Moller Institute, University of Southern Denmark, Odense, Denmark

* jguldager@health.sdu.dk

**Data Availability Statement:** All data files are available from the Zenodo database (DOI number: 10.5281/zenodo.7889353).

## Abstract

*VR FestLab* is a virtual reality party simulation application. The tool allows users to make decisions while experiencing a virtual party where they are offered alcohol. This study examines the user experience, game satisfaction, and engagement of 181 adolescent users (aged 15–18) with *VR FestLab* involving seven schools in Denmark. All user experience factors of the short user experience questionnaire were rated positively or neutral, and 66% of the students liked the VR experience. Neither the user experience score nor a score for game satisfaction and engagement were associated with sex, age, perceived family affluence, school performance, alcohol consumption and attitudes or mental health of students. Overall, positive user experiences and game satisfaction of *VR FestLab* were found not to differ according to student characteristics. We conclude that virtual simulations offer new ways for developing drinking refusal skills that are attractive and acceptable for adolescent users.

## 1. Introduction

Alcohol is a major risk factor for a number of diseases [1] and alcohol use among adolescents is a major public health concern. The highest rates of lifetime adolescent (aged 15–16) alcohol use was found in Czechia, Denmark and Hungary [2]. While 13% of European adolescents have experienced being drunk [2], higher rates were found in the eastern part of Europe with the highest (40%) in Denmark [2]. This is consistent with research showing that alcohol and smoking are the main explanation for the increase in social inequality in mortality since the 1980s in Denmark [3]. The high prevalence of alcohol use among European adolescents calls for new approaches for health promotion and alcohol prevention programs. A recent innovation in alcohol prevention is the newly developed virtual reality *(VR)* application *VR FestLab*,

**Funding:** This study is funded by a donation from TrygFonden, Denmark (Funding ID 129372, https://www.tryghed.dk/). The award was received by CS. The funder had no role in the study design or authority regards the conduct of the study and publication of results.

**Competing interests:** The authors have declared that no competing interests exist.

which is aimed at improving alcohol resistance skills among adolescents [4, 5]. VR was developed in the 1960s [6] and the usage heroff has increased significantly since in the past few years [7]. With the commercialization of VR systems and reduced hardware and software costs [7] it is now considered to be in its revival age [8]. But the increasing rise of VR can also be justified by the various application scenarios, such as for medical education [9]. Especially educational games taught through VR play an important role in offering new forms of interaction in virtual reality environments, making them even more attractive to users. Among the most important benefits offered by virtual reality are presence and telepresence, which refers to the feeling of being in an environment [10].

With this development, studies exploring the user experience of VR applications are only beginning to emerge and work needs to be expanded [8, 11]. Studying user experiences of VR is important for the design face of VR applications in general, where the focus should be not only on the product, process and design but also on the users and their behavior, interactions and emotions [12]. The usage of VR applications depends on the interactions with the user, and recent research has called for further studies on satisfaction and user experinces since perceptions of VR vary depending on the students and context [13]. Thus, an increased recognition of the potential role of individual factors on user experience are required to provide better experiences and to better understand the effects of VR on young people. Individual factors are the variation within user characteristics, such as demographic factors, psychological factors, domain-specific knowledge or prior experience which all may play a role [14]. Further, when developing health promotion interventions, to allow access to all individuals and reduce inequalities in health a longstanding attention has been given to reducing the differences based on sex, socioeconomic status and other social factors in the acces to these interventions [15]. Thus, it is essential to investigate if such individual differences exist in the UX and game satisfaction of VR applications for alcohol prevention, as such intervention should ideally appeal equally to all individuals.

While research on user experiences of VR applications is scarce, research is emerging. For example, studies looked at individual differences related to presence in VR experiences [16–19] and found that user experiences can be influenced by individual factors in many ways. For example, although a higher sense of presence has been reported for females in some studies [20], there is no consensus in the literature for sex differences in UX of VR experiences [21]. Moreover, some evidence exists which link cognitive abilities with a sense of presence in VR [22, 23], where the relationship between emotions or wellbeing and presences in VR has been contradictory [17, 21, 24]. Finally, qualitative data on the pilot study of *VR FestLab* indicated that students with different alcohol experiences expressed different views of *VR FestLab* [4] which is in line with previous research suggesting that different student populations will react differently to alcohol and drug education programs [25]. While some of those experienced with alcohol believed they could not learn so much from the game, this aspect was not mentioned by non-drinkers [4].

There is a need for more studies examining UX evaluations of VR applications in the area of prevention or health promotion [11], because UX is important for engagement. UX influences the effectiveness and sustainability of prevention tools and must be explored in the context of VR. A systematic review has found that the majority of UX evaluations of VR applications were conducted in the field of education and training, and highlighted that UX studies of VR applications which focus on children, and studies where the content is created with a 360˚ camera are needed [8].

*VR FestLab* is a simulation for adolescents which was co-created with adolescents by recordings of a party in a 360˚ format (see further at www.sdu.dk/vrfestlab). Although there is no opportunity for winning or losing in the simulation, it is experienced as a game [4]. We

conducted an interim evaluation of *VR Festlab* via a Randomised Control Trial indicating some positive outcome trends compared to control [5]. This article is focussed on better understanding user experiences on VR *Festlabs* attractiveness and the level of engagement that students experienced when playing the game. Specifically, this study aims to (a) examine the user experience, game satisfaction, and engagement of adolescents with the virtual simulation *VR FestLab*, and (b) to explore if sociodemographic factors, school performance, wellbeing factors or alcohol use and alcohol norms are associated with user experience, and game satisfaction and engagement ratings of the virtual reality simulation *VR FestLab*. The present study is explorative and a part of a larger research project analyzing the user experience and effectiveness of the VR tool [5].

## 2. Materials and methods

### 2.1 Study design

For this study, data on user experience, game satisfaction and engagement were collected in a cluster randomised control trial [5]. Schools were randomly assigned as clusters to the intervention or control group. While the students in the intervention schools played VR FestLab, the students in the control schools played another VR-based game (Oculus Quest—First Steps) [5]. Data for this study were from the intervention arm of the trial only and were collected before and immediately after the intervention (same in-class session) [5]. Data analyzed in this article is based on respondents from seven public or boarding schools in the region of Southern Denmark that were intervention schools in the trial. The data collection took place from August to December 2020 and from April to May 2021 with no data collection in between due to COVID-19 school closures.

### 2.2 Participants

To recruit the participants, the principals of schools in Southern Denmark were contacted by email. The recruitment itself was done by the principals or teachers in the schools themselves and all students who were between 15–18 years old were included, as the application was developed for this age group. Participants were excluded if they could not demonstrate fluency in Danish. In the cluster randomised controlled study [5], a sample size calculation was performed for the primary outcome Drinking refusal self-efficacy subscale. This was not done for the analyses of this study.

Data were collected through electronic questionnaires completed during school hours. The study adhered to Danish standards for ethical conduct of scientific studies and was approved by the Research Ethics Committee of the University of Southern Denmark on 18 December 2019 (case no 19/66794). The purpose of the study was explained to the students, and they were informed that data would be collected and presented in a completely anonymized form. In accordance with the Declaration of Helsinki, all students were informed that their participation was voluntary, and participants gave written consent (in Denmark, informed consent from parents is only required for children under 15 years).

### 2.3 The virtual simulation game VR FestLab

The alcohol prevention intervention *VR FestLab* is a virtual party simulation, where the players can direct their own virtual party experience through a series of choices. Throughout the simulation, the player is presented with different party scenarios and is offered a series of alcoholic and non-alcoholic beverages. The game was preinstalled on *Oculus Quest 1* VR devices and students were instructed in how to wear the device and navigate during the game with head

movements. Students tried *VR FestLab* for a maximum of 15 minutes in an in-class session followed by 45 minutes of structured reflection moderated by a trained study assistant.

## 2.4 Content of the questionnaires

The questionnaires were developed by the research team and pre-tested with 31 boarding school students [4]. The content of the questionnaire that was administered before the intervention session with playing the game covered the following areas:

**Sociodemographic information**: Participants were asked about their *sex* and *age*. With the question "How well off do you think you family is?" their *perceived family affluence* was measured on a five point scale (from not at all well-off to very well-off) based on the questionnaire of the Health Behavior in School-aged Children (*HBSC*) study [26]. For statistical analyses (Table 1) family affluence was categorized as "Low to medium" (Not at all well-off/Not so well-off/average) and "High" (Quite well-off/Very well-off).

**Binge drinking and injunctive norms regards alcohol**: *Binge drinking* was measured asking if the adolescent had ever been drinking five or more drinks on a single occasion. This question was based on the definition of binge drinking by the European School Survey Project on Alcohol and Other Drugs [27]. Further, questions regarding *injunctive norms* reflecting the attitudes towards permission to use alcohol were inspired by Pischke et al. [28], asking the adolescents which of five statements best described their attitudes towards using large amounts of

**Table 1. Characteristics of the study population (*n* = 181).**

|  | *n* | % |
|---|---|---|
| **Sex** |  |  |
| Male | 92 | 50.8 |
| Female | 89 | 49.2 |
| **Age** |  |  |
| 14 | 8 | 4.4 |
| 15 | 63 | 34.8 |
| 16 | 94 | 51.9 |
| 17 | 15 | 8.3 |
| 18 | <5 | 0.6 |
| **Perceived family affluence** |  |  |
| Low to medium | 160 | 88.4 |
| High | 21 | 11.6 |
| **Lifetime binge drinking** |  |  |
| No | 47 | 26.0 |
| Yes | 134 | 74.0 |
| **Injunctive norms** |  |  |
| Not OK to use alcohol for adolescents | 35 | 19.3 |
| OK to use alcohol for adolescents | 146 | 80.7 |
| **School performance** |  |  |
| Average or below | 73 | 39.8 |
| Good | 109 | 60.2 |
| **Mental health (SDQ score)** |  |  |
| Normal | 139 | 76.8 |
| Slightly elevated difficulties | 16 | 8.9 |
| High difficulties | 10 | 5.6 |
| Very high difficulties | 16 | 8.8 |

alcohol so one becomes drunk 1) "Is never a good idea–no matter how old you are", 2) "Is at my age not okay to use but okay if you are an adult", 3) "Is at my age okay to use occasionally if it doesn't interfere with study or other obligations", 4) "Is at my age okay to use occasionally even if it does interfere with study or work", 5) "Is at my age okay to use frequently if that is what the person wants to do"). For the analysis, agreement with statement 1) and 2) were categorized as "Not OK to use for adolescents", and agreement with statement 3), 4) or 5) as "OK to use alcohol for adolescents".

**School performance and mental health**: Questions concerning *school performance* were adapted from the HBSC study [26] asking, compared to classmates, how the adolescent believed his/her teacher was rating his/her school performance (below average/average/good/very good). For the analysis the categories "very good" and "good" were categorized as "High school performance" and the remaining categories were transferred into "medium to low school performance". We also applied the 25 items of the Strengths and Difficulties Questionnaire (SDQ) in its Danish version to establish the *mental health* of the students [29]. A three point Likert scale form "not true" to "certainly true" was used. The Cronbach´s alpha of the SDQ scale in our sample was 0.79. The items were categorized into "total difficulties" and categorized to reflect the Danish norms into normal (0–14), slightly elevated (15–17), high (18–20), and very high (21–40) level of difficulties [30] for statistical analyses (Table 1).

Directly after *VR FestLab* was tested the second questionnaire was administered to the students which contained the following scales:

**User experience**: *User experiences* of *VR FestLab* was measured using the short version of the User Experience Questionnaire-Short (UEQ-S). By using bipolar adjectives *VR FestLab* was rated using a 7-point Likert scale in eight items: 1) obstructive/supportive, 2) complicated/easy, 3) inefficient/efficient, 4) confusing/clear, 5) boring/exciting, 6) not interesting/interesting, 7) conventional/inventive, and 8) usual/leading edge [31]. According to the authors [32] items one to four represented the pragmatic quality, and items five to eight represented hedonic quality. Values between -0.8 and 0.8 represent a neutral evaluation, $> 0.8$ a positive evaluation, and $< -0.8$ a negative evaluation. The Cronbach´s alpha of the Short EUQ was 0.87 in our sample.

**Game satisfaction and engagement**: To establish *game satisfaction and engagement*, we asked the participants to rate the following statements on a 5-point Likert scale from "disagree a lot" to "agree a lot": "I could engage in the game", "I would like to explore the game further", "I liked the VR experience", "The party situation is realistic", "I would recommend the game to my friends", "The characters in the game are realistic","I liked the music" on a five point Likert Scale (disagree a lot, disagree to some degree, don't know, agree to some degree, and agree a lot). A sum score of the six items was created for the analysis. Cronbach´s alpha of the Short EUQ was 0.85 in our sample.

## 2.5 Statistical analysis

Participants´ characteristics are reported as absolute and relative frequencies in Table 1. Mean values of UEQ items are reported in Fig 1 and frequencies of participants responses on game satisfaction and engagement items in Fig 2. Bivariate linear regression analyses were used to study if the selected participants´ characteristics were associated with the total UEQ-S score and the UEQ-S sub-scores pragmatic and hedonistic quality (Table 2). It was further investigated if the same set of variables were associated with satisfaction and engagement score as dependent variables (Table 3). Results are presented as estimates (β) with 95% confidence intervals. The Statistical Package for the Social Science 28 (SPSS) for Windows was used to conduct statistical analysis.

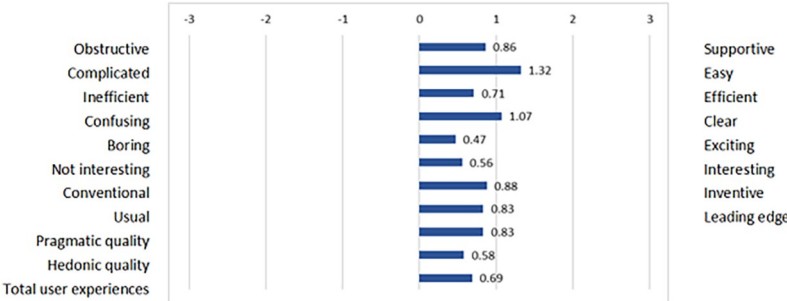

*: Rated from fully agree with negative term (-3) to fully agree with positive term (+3).

**Fig 1. Mean ratings of the single items of the UEQ, and mean scores for pragmatic and hedonic quality ($n$ = 128–179)*.**

## 3. Results

In total 183 of 217 students (response rate 84.3%) completed the first and 181 the second questionnaire resulting in an attrition rate of only 16.5%.

### 3.1 Description of the sample

Table 1 depicts information about the characteristics of the study population. The students were evenly distributed regarding sex (49% female), the majority (87%) was 15-16-years-old, and the most (88%) had low to medium family affluence. Regarding the use of alcohol, 74% admitted to having experimented with binge drinking at least once, and 81% believed it was OK for adolescents to use alcohol. Sixty percent stated that their performance in school was high, and 77% identified as having normal mental health.

### 3.2 User experience

The student responses on user experiences of *VR FestLab* are presented in Fig 1. Pragmatic quality with a mean of 0.83 (SD 1.17) can be categorized as positive rating, and total user experiences with a mean of 0.69 (SD 1.16) and hedonic quality with a mean of 0.58 (SD 1.22) as neutral in our sample according to the established cut-offs. Further items evaluated as positive were: complicated/easy (mean 1.30, SD 1.71), confusing/clear (mean 1.07, SD 1.60), conventional/inventive (mean 0.88, SD 1.52), obstructive/supportive (mean 0.86, SD 1.18), and usual/leading edge

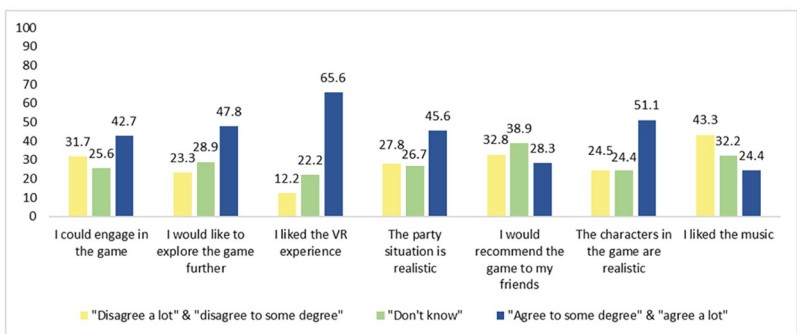

**Fig 2. Game satisfaction and engagement of the study population (in percentages) ($n$ = 180).**

**Table 2. Bivariate linear regressions for factors associated with pragmatic and hedonic quality of user experience.**

| | Pragmatic quality score | | | Hedonic quality score | | | Total user experiences score | | |
| | N = 114* | | | N = 141* | | | N = 100* | | |
| | Beta | (95% CI) | p-value | Beta | (95% CI) | p-value | Beta | (95% CI) | p-value |
|---|---|---|---|---|---|---|---|---|---|
| **Sex** (male = 0, female = 1) | -0.09 | -0.53–0.35 | 0.68 | 0.01 | -0.40–0.42 | 0.96 | 0.08 | -0.39–0.54 | 0.74 |
| **Age** | -0.14 | -0.44–0.16 | 0.35 | -0.03 | -0.31–0.25 | 0.81 | -0.08 | -0.39–0.23 | 0.60 |
| **Perceived family affluence** (1–5) | 0.20 | -0.39–0.80 | 0.50 | 0.01 | -0.53–0.54 | 0.98 | 0.22 | -0.41–0.85 | 0.50 |
| **Lifetime binge drinking** (no = 0, yes = 1) | -0.18 | -0.68–0.32 | 0.47 | 0.08 | -0.38–0.53 | 0.75 | -0.02 | -0.55–0.50 | 0.93 |
| **Injunctive norms for alcohol** (Not OK to use = 0, OK to use alcohol = 1) | -0.09 | -0.66–0.47 | 0.74 | 0.06 | -0.46–0.57 | 0.83 | -0.04 | -0.64–0.56 | 0.90 |
| **School performance** (1–4) | 0.22 | -0.08–0.53 | 0.15 | 0.19 | -0.09–0.47 | 0.19 | 0.21 | -0.10–0.52 | 0.18 |
| **Mental health (SDQ) score** (0–40) | -0.02 | -0.06–0.02 | 0.40 | -0.02 | -0.05–0.02 | 0.42 | -0.20 | -0.06–0.02 | 0.37 |

*: The sample included participants with valid scores on all variables for pragmatic quality, hedonic quality, and total user experiences, respectively.

(mean 0.83, SD 1.42). Neutrally evaluated items were: inefficient/efficient (mean 0.71, SD 1.38), not interesting/interesting (mean 0.56, SD 1.46), and boring/exciting (mean 0.47, SD 1.43).

## 3.3 Game satisfaction and engagement

Percentages of participants agreeing with the game satisfaction and engagement items are presented in Fig 2. Most students were positive regarding the overall satisfaction with the game and liked the VR experience (66%). They also regarded the characters in the game as realistic (51%). However, only one quarter was satisfied with the music in the game (24%). The average game satisfaction and engagement of the seven variables combined was 3.2 (SD 0.81) on a scale from 0–5.

## 3.4 Factors associated with user experience

The bivariate linear regression analyses showed that none of the variables in the regression model was significantly associated with the pragmatic quality score, the hedonic quality score or the total user experiences score of *VR FestLab*.

## 3.5 Factors associated with game satisfaction and engagement

A bivariate linear regression analysis showed that none of the variables in the regression model was significantly associated with the game satisfaction and engagement score of *VR FestLab*.

**Table 3. Bivariate linear regressions for factors associated with the game satisfaction and engagement score (n = 180).**

| | Game satisfaction and engagement score | | |
| | Beta | (95% CI) | p-value |
|---|---|---|---|
| **Sex** (male = 0, female = 1) | 0.12 | -0.12–0.36 | 0.34 |
| **Age** | -0.06 | -0.23–0.10 | 0.45 |
| **Perceived family affluence** (1–5) | -0.04 | -0.34–0.26 | 0.79 |
| **Lifetime binge drinking** (no = 0, yes = 1) | -0.03 | -0.31–0.24 | 0.81 |
| **Injunctive norms for alcohol** (Not OK to use = 0, OK to use alcohol = 1) | -0.06 | -0.37–0.24 | 0.68 |
| **School performance** (1–4) | -0.00 | -0.17–0.16 | 0.98 |
| **Mental health (SDQ) score** (0–40) | 0.02 | -0.00–0.04 | 0.10 |

## 4. Discussion

The present study investigated the user experiences, game satisfaction, and engagement of adolescents with the virtual reality simulation *VR FestLab*. We found that most user experience factors of the user experience questionnaire were rated positively and some neutral, while no negative rating occurred. Compared to other studies evaluating user experiences in general by using the Short UEQ, our rating of the overall score for attractiveness is slightly below average, while several of the single ratings regards ease of use, clarity, innovative and leading edge aspect of the game were rated above average [32]. However, it should be mentioned, that these studies were based on usability of web pages, software, web shops, and social networks which can be difficult to compare to a VR application. In the area of health-related VR applications, existing studies are based on very different areas than ours (such as fall reduction in elderly [33] or safety of driving [34, 35]). However, a recent study on the usage of VR to engage adolescents in physical activity showed that adolescents saw great potential in VR to engage in physical activity [36]. Another study [37] in VR-based smoking prevention that dealt with the acceptance of VR-based learning games in schools showed that young people accept these kinds of learning games. The authors [37] stated that especially the gaming character of VR promotes engagement and learning motivation for preventive behaviours. These results align with our results, as most of the participants positively evaluated the engagement in the game, the VR experience and the further exploration of VR FestLab. It is therefore possible that VR with its immersive character is a promising approach to embed the topic of substance use prevention among students. Whether VR FestLab with its engagement elements and the VR experience also leads to behavioural change in terms of rational alcohol consumption and the reduction of alcohol-related risk behaviour should be clarified in further studies.

We found that most students had positive game satisfaction regards most of the aspects that were rated with the exemption of the music in the game that some respondents did not like. Such a generally positive game satisfaction is important, as it has been described that users´ interaction and learning experiences of virtual reality has an impact on the learning outcome hereof [38]. VR has demonstrated promising effects for smoking, nutrition, physical activity, and obesity [11] and game satisfaction is an important component for effective VR interventions.

Interestingly, none of the included factors (sex, age, perceived family affluence, school performance, alcohol consumption and attitudes and mental health of students) were related to game satisfaction and engagement of *VR FestLab* or to any of the user experience questionnaire score of the sub-scores for hedonic quality and pragmatic quality [39]. This indicates that the game is equally attractive to students regardless of the personal characteristics of adolescents. This is important in a health promotion perspective, because sometimes health promoting programmes are better rated and accepted by girls than by boys [40, 41]. Since there has been a long-standing focus on reducing differences in health based on sex, socioeconomic status and other social factors [15], developing interventions that are acceptable to users from all social strata is an important step towards this goal. However, more research is needed to study if VR FestLab is also effective among diverse user groups because previous research has identified that the effects of an online school based alcohol education program to some extent differered for subgroups regarding drinking experience [25].

Overall, our findings contribute to the knowledge base of how to best approach and reach adolescents as target group for digital alcohol prevention. We can conclude that the virtual simulation game is acceptable and attractive for adolescent users. This is an important finding,

since a recent review pointed out that more research on the use of VR interventions on alcohol consumption and on younger age groups is needed [11].

## 4.1 Limitations

The results of our research should be interpreted in the light of the limitations of the study. Firstly, despite our sample size being larger than in related studies (for example [18, 20, 21], it might not have been sufficient to identify an association between the studied factors and user experience of VR FestLab in the regression model.

Data are based on the experiences of students from seven Danish schools located in the Region Southern Denmark. It is unknown if the results would differ, if more schools from several regions would have been included. Further, it cannot be ruled out that teachers of the participating school classes may have been more positive towards new technologies since they accepted participation in the study, which could have affected the students' ratings of UX. However, since such a selection bias was on teacher and not student level the impact on student ratings should be limited.

In the same line, it cannot be ruled out that ratings can be biased due to self-reports of the students in terms of potential selective memory and/or positive exaggeration of gameplay satisfaction and engagement [42]. However, to limit this bias, data were collected on the same day the students tried VR FestLab. It should be noticed, that with gameplay being limited to 15 minutes some students may not have encountered all game possibilities, which could have affected their user experiences, and this study did not include data on which scenes each participating student were not exposed to.

Finally, in our study we employed a specific VR technology (Oculus Quest 1). VR technology is emerging rapidly, and our results may not be generalized to other more modern types of VR technology. In addition, we cannot report any user experiences from longer term use of VR FestLab.

## 5. Conclusions

This study aimed at investigating user experiences, game satisfaction, and engagement of adolescents with the virtual reality simulation VR FestLab, an innovative alcohol prevention tool designed for Danish students aged 15–18. Our study found that user experiences were rated positively or neutral, and that most students had positive game satisfaction. No negative experiences were reported. None of the student characteristics studied (sex, age, perceived family affluence, school performance, alcohol consumption and attitudes and mental health of students) were associated with user experience or with game satisfaction and engagement. We conclude that virtual simulations offer new ways for developing drinking refusal skills that are attractive and acceptable for adolescent users. VR is a promising alcohol education tool that warrants future research application and evaluation so we can better assess its role in health education programs.

Further research is needed to build upon these results and explore additional factors that may influence user experiences with VR simulations in prevention. Further, it would be relevant to investigate user experiences after a longer duration of gameplay. Finally, research is needed linking exposure/dose received of such VR applications by e.g. tracking user decisions during gameplay with aspects of user experiences.

Future work could focus on further development of the usage of VR in health promotion, as to date, very few applications and very little research has been devoted to this area of health services. We found that the users experienced the VR application positively and that the game was equally attractive to students regardless of the personal characteristics of the adolescents.

Thus, as technologies rapidly emerge, the usage of VR could prove very valuable in the area of health promotion for adolescents.

Finally, as alcohol use among adolescents is a major public health concern, this calls for new innovative school based alcohol prevention programs. These efforts may be enriched with the usage of new techologies like VR applications which are appealing to the youth and can offer a new approach to teaching health promotion.

## Acknowledgments

We would like to extend a warm thank you to the participating students, their teachers and the school principals for their participation in this project. Also, we would like to thank Laura Nørgaard Lehnsted and Maria Busk Damgaard, MScPH for their contributions to the field work.

## Author Contributions

**Conceptualization:** Julie Dalgaard Guldager, Timo Dietrich, Christiane Stock.

**Data curation:** Julie Dalgaard Guldager.

**Formal analysis:** Julie Dalgaard Guldager.

**Funding acquisition:** Christiane Stock.

**Investigation:** Julie Dalgaard Guldager, Christiane Stock.

**Methodology:** Julie Dalgaard Guldager, Christiane Stock.

**Project administration:** Julie Dalgaard Guldager, Christiane Stock.

**Supervision:** Christiane Stock.

**Validation:** Christiane Stock.

**Writing – original draft:** Julie Dalgaard Guldager.

**Writing – review & editing:** Julie Dalgaard Guldager, Robert Hrynyschyn, Satayesh Lavasani Kjær, Timo Dietrich, Gunver Majgaard, Christiane Stock.

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
