## [Decision Letter · Decision Letter 0]

24 Apr 2023

PONE-D-23-03374User experience, game satisfaction and engagement with the virtual simulation VR FestLab for alcohol prevention: A quantitative analysis among Danish adolescentsPLOS ONE

Dear Dr. Guldager,

Thank you for submitting your manuscript to PLOS ONE. After careful consideration, we feel that it has merit but does not fully meet PLOS ONE’s publication criteria as it currently stands. Therefore, we invite you to submit a revised version of the manuscript that addresses the points raised during the review process.

We look forward to receiving your revised manuscript.

Kind regards,

Ali Garavand

Academic Editor

PLOS ONE

Reviewers' comments:

Reviewer's Responses to Questions

**Comments to the Author**

1. Is the manuscript technically sound, and do the data support the conclusions?

Reviewer #1: Yes

Reviewer #2: Yes

2. Has the statistical analysis been performed appropriately and rigorously? 

Reviewer #1: Yes

Reviewer #2: Yes

3. Have the authors made all data underlying the findings in their manuscript fully available?

Reviewer #1: Yes

Reviewer #2: Yes

4. Is the manuscript presented in an intelligible fashion and written in standard English?

Reviewer #1: Yes

Reviewer #2: Yes

5. Review Comments to the Author

Reviewer #1: The novelty and contribution of this paper is “there”, but some changes are needed. Here are my comments on improving the manuscript:

1. Overall:

a) Why do the authors conduct this study? The research contributions are weak. Please kindly explain.

b) Please consider how to effectively integrate some review papers and update.

2. Introduction:

a) Research questions, that drive the paper, should be built in the introduction from an ongoing and pertinent bibliography (up to 2022-23) and these should be of global interest and not focused on a particular local problem. Identifying a research gap is the most important by indicating in-text some newer references that are significant to your particular field of research.

3. Discussion:

a) Authors should answer your research question in the conclusions and discussion. Please provide a reasonable need to read your work’s results than previous ones or simply answer what we learned compared with current, significant research (up to 2022 should be your work’s “significance”).

b) Are there any points of view related to the consequences of this study’s limitations that may have an impact on their findings?

4. Conclusions and limits are too short for such a study.

a) Are there any points of view related to the consequences of this study’s limitations that may have an impact on their findings?

b) Implications for practice and method are not provided.

Minor comment: Please search if any other references in-text need reconstruction, e.g., (Tatnell et al., 2022) -> [9], see line 235.

Reviewer #2: Firstly, I would like to thank you for this research.

In general, the manuscript is written with an easy-to-follow and readable layout. However, the manuscript needs some modifications including:

In the abstract section, keywords should be based on the Mesh terms.

More and newer references should be used in the introduction. Only one reference is mentioned in several paragraphs.

The sampling method is not clearly presented in the method section and sample size is not big enough.

The wider discussion on the application of virtual simulation in preventive behaviors in schools would add value to the overall manuscript. Compare the results of the study with similar studies and check the reasons for the difference.

References need to be updated.

6. PLOS authors have the option to publish the peer review history of their article (what does this mean?). If published, this will include your full peer review and any attached files.

Reviewer #1: No

Reviewer #2: No

---

## [Author Response · Author response to Decision Letter 0]

5 May 2023

Manuscript ID: PONE-D-23-03374

Title: User experience, game satisfaction and engagement with the virtual simulation VR FestLab for alcohol prevention: A quantitative analysis among Danish adolescents

Dear reviewer,

We would like to thank you for your valuable feedback and sincerely appreciate your comments in order to improve our manuscript. We have addressed each concern and changed the manuscript accordingly. 

Reply to reviewer 1’s comments:

a) Why do the authors conduct this study? The research contributions are weak. Please kindly explain.

Response: Thank you for pointing out that this has not been explained in detail. We have added several elaborations throughout the introduction, and further explained the rationale behind conducting the study by adding these paragraphs:

Further, when developing health promotion interventions, to allow access to all individuals and reduce inequalities in health a longstanding attention has been given to reducing the differences based on sex, socioeconomic status and other social factors in the acces to these interventions [11]. Thus, it is essential to investigate if such individual differences exist in the UX and game satisfaction of VR applications for alcohol prevention, as such intervention should ideally appeal equally to all individuals. (lines 64-69)

Studying user experiences of VR is important for the design face of VR applications in general, where the focus should be not only on the product, process and design but also on the users and their behavior, interactions and emotions [12]. The usage of VR applications depends on the interactions with the user, and recent research has called for further studies on satisfaction and user experinces since perceptions of VR vary depending on the students and context [13]. (lines 55-59)

b) Please consider how to effectively integrate some review papers and update.

Response: As the research in this area is scarce, we have not been able to identify several relevant review papers. However, we have integrated the review by Barteit et al.

2. Introduction: Research questions, that drive the paper, should be built in the introduction from an ongoing and pertinent bibliography (up to 2022-23) and these should be of global interest and not focused on a particular local problem. Identifying a research gap is the most important by indicating in-text some newer references that are significant to your particular field of research.

Response: To more clearly indicate the research gaps, we have elaborated upon this in the introduction and have added research from newer references (Barteit S, et al. 2021, Mütterlein, J. 2018, Auernhammer, J. 2020, Mustafa, B. 2022, Grassini, S. et al. 2021, Kober, S. E., & Neuper, C. 2013). Further, regarding alcohol use, we have altered the focus from being on Denmark to being on Europe in general. We have added:

The highest rates of lifetime adolescent (aged 15-16) alcohol use is found in Czechia, Denmark and Hungary [2]. While 13% of European adolescents have experienced being drunk [2], the higher rates are found in the eastern part of Europe with the highest (40%) in Denmark [2]. (lines 39-41

3. Discussion: a) Authors should answer your research question in the conclusions and discussion. Please provide a reasonable need to read your work’s results than previous ones or simply answer what we learned compared with current, significant research (up to 2022 should be your work’s “significance”).

Response: The research question concerns the examination of the user experience and game satisfaction, and an exploration if several factors are associated where these two concepts. In the present conclusion and discussion, we have stated how user experiences and game satisfaction were rated, and that none of the student characteristics studies were associated with these two concept.Furthermore, we have added a paragraph discussing what we have learned compared to current research in more detail. We have added:

Another study [37] in VR-based smoking prevention that dealt with the acceptance of VR-based learning games in schools showed that young people accept these kinds of learning games. The authors [37] stated that especially the gaming character of VR promotes engagement and learning motivation for preventive behaviours. These results align with our results, as most of the participants positively evaluated the engagement in the game, the VR experience and the further exploration of VR FestLab. It is therefore possible that VR with its immersive character is a promising approach to embed the topic of substance use prevention among students. Whether VR FestLab with its engagement elements and the VR experience also leads to behavioural change in terms of rational alcohol consumption and the reduction of alcohol-related risk behaviour should be clarified in further studies. (lines 255-263)

4. Conclusions and limits are too short for such a study.

a) Are there any points of view related to the consequences of this study’s limitations that may have an impact on their findings?

Response: To elaborate on the consequences of the limitations of the study, we have added two aspects regarding the limitations (sample size and type of VR technology used), and specified in which way the self-reporting of the students can have affected the findings.

We have added these paragraphs:

Firstly, despite our sample size being larger than in related studies (for example [18,20,21], it might not have been sufficient to identify an association between the studied factors and user experience of VR FestLab in the regression model.(lines 289-291)

Finally, in our study we employed a specific VR technology (Oculus Quest). VR technology is emerging rapidly, and our results may not be generalized to other more modern types of VR technology.(lines 306-308)

b) Implications for practice and method are not provided.

Response: Thank you for pointing out, that implications for practice have not been provided (implications for method/research is described in the end of the conclusion. We have added this paragraph:

Future work could focus on further development of the usage of VR in health promotion, as to date, very few applications and very little research has been devoted to this area of health services. We found that the users experienced the VR application positively and that the game was equally attractive to students regardless of the personal characteristics of the adolescents. Thus, as technologies rapidly emerge, the usage of VR could prove very valuable in the area of health promotion for adolescents.

Finally, as alcohol use among adolescents is a major public health concern, this calls for new innovative school based alcohol prevention programs. These efforts may be enriched with the usage of new techologies like VR applications which are appealing to the youth and can offer a new approach to teaching health promotion. (lines 324-333)

Minor comment: Please search if any other references in-text need reconstruction, e.g., (Tatnell et al., 2022) -> [9], see line 235.

Response: This has been checked for and updated.

Reply to reviewer 2’s comments:

In the abstract section, keywords should be based on the Mesh terms.

Response: Thank you for pointing this out. We have added relevant MeSH terms to the key words.

More and newer references should be used in the introduction. Only one reference is mentioned in several paragraphs.

Response: We have added a paragraph to better present the role of VR and its application scenarios and have introduced more references: 

But the increasing rise of VR can also be justified by the various application scenarios, such as for medical education [9]. Especially educational games taught through VR play an important role in offering new forms of interaction in virtual reality environments, making them even more attractive to users. Among the most important benefits offered by virtual reality are presence and telepresence, which refers to the feeling of being in an environment [10]. (lines 48-53)

The sampling method is not clearly presented in the method section and sample size is not big enough.

Response: Thank you very much for the advice. We have now added two sections on sampling methods and sample size calculation to make this more transparent, and we added a section on the consequences hereof to the limitation section: 

For this study, data on user experience, game satisfaction and engagement were collected in a cluster randomised control trial [5]. Schools were randomly assigned as clusters to the intervention or control group. While the students in the intervention schools played VR FestLab, the students in the control schools played another VR-based game (Oculus Quest—First Steps) [5]. (lines 102-105)

To recruit the participants, the principals of schools in Southern Denmark were contacted by email. The recruitment itself was done by the principals or teachers in the schools themselves and all students who were between 15-18 years old were included, as the application was developed for this age group. Participants were excluded if they could not demonstrate fluency in Danish. In the cluster randomised controlled study [5], a sample size calculation was performed for the primary outcome Drinking refusal self-efficacy subscale. This was not done for the analyses of this study.(lines 112-118)

Firstly, despite our sample size being larger than in related studies (for example [18,20,21], it might not have been sufficient to identify an association between the studied factors and user experience of VR FestLab in the regression model. (lines 289-291)

The wider discussion on the application of virtual simulation in preventive behaviors in schools would add value to the overall manuscript. Compare the results of the study with similar studies and check the reasons for the difference.

Response: Thank you for pointing this out. We have added the following section to relate our results more closely to other studies:

Another study [37] in VR-based smoking prevention that dealt with the acceptance of VR-based learning games in schools showed that young people accept these kinds of learning games. The authors [37] stated that especially the gaming character of VR promotes engagement and learning motivation for preventive behaviours. These results align with our results, as most of the participants positively evaluated the engagement in the game, the VR experience and the further exploration of VR FestLab. It is therefore possible that VR with its immersive character is a promising approach to embed the topic of substance use prevention among students. Whether VR FestLab with its engagement elements and the VR experience also leads to learning success in terms of rational alcohol consumption and the reduction of alcohol-related risk behaviour should be clarified in further studies. (lines 255-263)

References need to be updated.

Response: References has been updated. For example, we have added these references: Barteit S, et al. 2021, Mütterlein, J. 2018, Auernhammer, J. 2020, Mustafa, B. 2022, Grassini, S. et al. 2021, Kober, S. E., & Neuper, C. 2013.

---

## [Decision Letter · Decision Letter 1]

18 May 2023

User experience, game satisfaction and engagement with the virtual simulation VR FestLab for alcohol prevention: A quantitative analysis among Danish adolescents

PONE-D-23-03374R1

Dear Dr. Guldager

We’re pleased to inform you that your manuscript has been judged scientifically suitable for publication and will be formally accepted for publication once it meets all outstanding technical requirements.

Kind regards,

Ali Garavand

Academic Editor

PLOS ONE

Reviewers' comments:

Reviewer's Responses to Questions

**Comments to the Author**

1. If the authors have adequately addressed your comments raised in a previous round of review and you feel that this manuscript is now acceptable for publication, you may indicate that here to bypass the “Comments to the Author” section, enter your conflict of interest statement in the “Confidential to Editor” section, and submit your "Accept" recommendation.

Reviewer #1: All comments have been addressed

Reviewer #2: (No Response)

2. Is the manuscript technically sound, and do the data support the conclusions?

Reviewer #1: Yes

Reviewer #2: (No Response)

3. Has the statistical analysis been performed appropriately and rigorously? 

Reviewer #1: Yes

Reviewer #2: (No Response)

4. Have the authors made all data underlying the findings in their manuscript fully available?

Reviewer #1: Yes

Reviewer #2: (No Response)

5. Is the manuscript presented in an intelligible fashion and written in standard English?

Reviewer #1: Yes

Reviewer #2: (No Response)

6. Review Comments to the Author

Reviewer #1: The authors have satisfactorily responded to all my questions and made the necessary changes to the manuscript.

Reviewer #2: (No Response)

7. PLOS authors have the option to publish the peer review history of their article (what does this mean?). If published, this will include your full peer review and any attached files.

Reviewer #1: No

Reviewer #2: No

---

## [Editor Report · Acceptance letter]

19 May 2023

PONE-D-23-03374R1 

User experience, game satisfaction and engagement with the virtual simulation VR FestLab for alcohol prevention: A quantitative analysis among Danish adolescents 

Dear Dr. Guldager:

I'm pleased to inform you that your manuscript has been deemed suitable for publication in PLOS ONE. Congratulations! Your manuscript is now with our production department. 

Kind regards, 

on behalf of

Dr. Ali Garavand 

Academic Editor

PLOS ONE